# Physical harms in colorectal cancer screening: An overview of the reporting in systematic reviews and randomised controlled trials

Anne Katrine Lykke Bie[ID][1]*, Frederik Handberg Juul Martiny[1,2],
Christian Patrick Jauernik[1☯], Or Joseph Rahbek[1☯], Sigrid Brisson Nielsen[1☯],
Emma Grundtvig Gram[ID][1,3☯], Isabella Kindt[1☯], John Brandt Brodersen[1,3,4]

1 Section of General Practice and Research Unit for General Practice, Department of Public Health, University of Copenhagen, Copenhagen, Denmark, 2 Department of Social Medicine, Bispebjerg and Frederiksberg Hospital, Copenhagen, Denmark, 3 Research Unit for General Practice in Region Zealand, Copenhagen, Denmark, 4 Research Unit for General Practice, Department of Community Medicine, Faculty of Health Sciences, UiT The Arctic University of Norway, Tromsø, Norway

☯ These authors contributed equally to this work.
* aklykkebie@gmail.com

## Abstract

### Objective

To assess the comprehensiveness of the reporting of physical harms in colorectal cancer screening programmes (CRCSPs) in randomised controlled trials (RCTs) and systematic reviews (SRs).

### Design

We conducted an overview of reviews, comparing the comprehensiveness of reporting of harms in SRs and RCTs with a recent SR conducted according to the PRISMA-harms extension, identifying 17 types of physical harm potentially resulting from CRCSPs.

### Main outcome measures

Proportion of the 17 types of physical harm reported per study (study coverage), across studies (outcome coverage) and the level of harm severity reported in RCTs and SRs.

### Results

We identified 24 RCTs and 16 SRs investigating physical harms related to CRCSPs. The median study coverage was 4 and 3 out of the 17 types of harm, varying from 5.9–47.1% and 5.9–52.9% types of physical harm reported in RCTs and SRs, respectively. The median outcome coverage was 4 and 3 across RCTs and SRs, varying from 0–66.7% and 0–87.5% in RCTs and SRs, respectively. Of note, 4 types of harm

**Data availability statement:** All data is available in the attached manuscript, either in the full text or appendices.

**Funding:** This work was supported by the Lundbeck Foundation (AKB) and "Sara Krabbes Legat" (FM), the Danish Society for General Practitioners (FM) and via the Danish Cancer Society (FM). The funders had no role in study design, data collection and analysis, decision to publish, or preparation of the manuscript. The first author is independent of the funding bodies.

**Competing interests:** The authors have declared that no competing interests exist.

were not reported in any of the identified SRs. Inconsistent definitions of harm in RCTs and SRs made it difficult to assess which levels of severity of harm that were reported in studies.

## Discussion

Poor reporting of harms in RCTs was compounded in SRs. We found poor study and outcome coverage and considerable inconsistencies concerning how physical harms were defined in RCTs and SRs. The inconsistent reporting of harms may result in an underestimation of their magnitude in relation to CRCSPs, raising concerns about our current capacity to evaluate the safety of these programmes. Adequate use of existing guidelines for harm reporting in RCTs and SRs and international consensus on how best to define and measure harms in studies of CRCSPs is warranted.

## Introduction

Colorectal cancer (CRC) is the third most common cancer worldwide and the second largest contributor to cancer-related deaths globally [1]. In many countries in Europe, North- and South America, Asia, and Oceania, screening is recommended to reduce incidence and mortality from colorectal cancer [2]. According to Wilson and Jungner's principles and practice of screening [3], the benefits and harms of screening should be assessed before implementing a screening programme. In addition, clinical guidelines regulating practical aspects of screening, i.e., who to screen, how to screen, the time between screening rounds, etc., must be developed, ideally by a group of multidisciplinary experts, supported by the best available evidence [4]. Such a guideline regarding CRCSPs has recently been carried out [5].

Clinical guidelines should rely on well-conducted and well-reported RCTs and SRs. This is particularly important when it concerns potential benefits and harms of medical interventions, e.g., screening [6], since many seemingly healthy people are exposed to screening, and thus risk being harmed. In 2016, the PRISMA-harms extension to the PRISMA statement was developed to improve the conduct and reporting of harms from medical interventions in SRs.

The motive for developing the extension was that SRs tend to compound poor reporting of harms in primary studies [7]. The counterpart to this guideline for RCTs is the CONSORT-harms extension [8] guidance on reporting harms in RCTs. However, we have found that this reporting guideline is seldom used in RCTs investigating CRCSPs [9]. Since the PRISMA-harms extension has been published quite late compared to the evidence base of SRs about CRCSPS, and the CONSORT-harms guideline is seldom used for RCTs, it is not surprising that the harms of screening, in general, tend to be heterogeneously and scarcely reported [10–12].

Due to these concerns about the trustworthiness of the reporting of harms related to CRCSPs, a systematic review was conducted, assessing the best available evidence on physical harm from CRCSPs. Here, 17 types of physical harm potentially resulting from CRCSPs [9] were identified. Moreover, the authors found a

surprising number of studies not formerly included in existing SRs, and significant heterogeneity concerning how harms were defined, whether certain types of harm were assessed in studies, how harms were measured, and how they were reported. In addition, the authors found many shortcomings in existing SRs concerning scrutiny of the methodological quality of studies and concerning their quality of harm reporting compared to our findings [9], underpinning the rationale for the PRISMA-harms extension.

However, even though the findings pointed towards important methodological issues related to the reporting of physical harms related to CRCSPs, it was not designed to systematically address the issues stumbled upon. Therefore, the authors decided that it was warranted to explore the reporting quality of physical harm in RCTs and SRs further in an overview of reviews. Therefore, we conducted the present study, aiming to assess the comprehensiveness of the reporting of physical harm in publications from RCTs and SRs investigating CRCSPs using the 17 types of physical harm as a reference standard [9].

Fig 1 illustrates the screening cascade of CRCSPs. The reference review and this overview of RCTs and SRs focus on the physical harms that may occur during the diagnostic workup phase of the CRCSP, which includes the endoscopic procedures and their related procedures, e.g., bowel preparation, anaesthesia, etc.

## Methods

The methodological basis for this study is the PRISMA-HARMS checklist [13], the PRIOR statement [14], a 54-item checklist concerning reporting of methods in overviews of reviews [15] and guidance from the Cochrane Collaboration [16]. For further details on the review process, please view recently published articles from the systematic review, from here on referred to as *the reference review* [9,17]. We did not develop a separate protocol for this overview of reviews and RCTs because the need for the overview, i.e., the poor reporting of harms in RCTs and systematic reviews, became apparent during the conduct of the reference review.

### 17 types of physical harm as a reference standard

The reference review identified each distinct harm, e.g., bleeding, pain and perforation, reported in the included studies. From these harms, we sorted harms into 17 *types* of physical harm: Death, Perforation, Cardiovascular- and pulmonary complications, Bleeding, Post-polypectomy syndrome, Infections, Inflammatory complications, Colorectal symptoms,

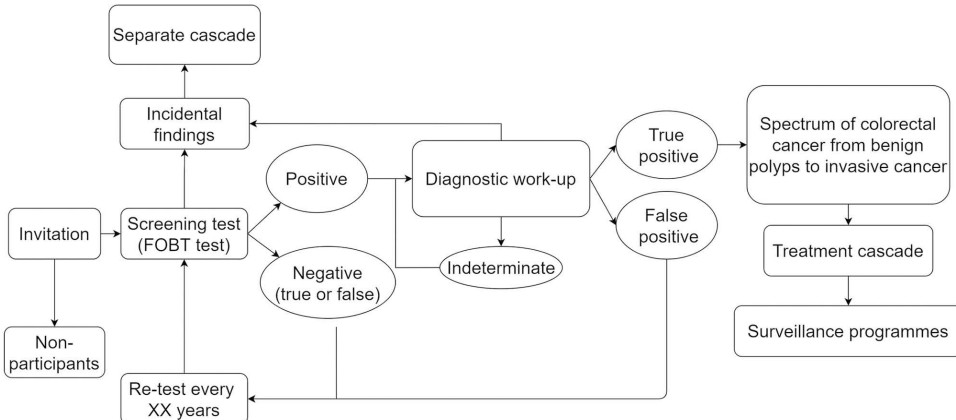

**Fig 1. Colorectal cancer screening cascade.** The figure illustrates the screening cascade, from the receipt of invitation letters through diagnosis, treatment, and/or entry into surveillance programmes.

Sedation-related complications, Complications related to bowel preparation, Sleep disturbances, Nausea/vomiting, Dizziness, Pain, Discomfort, Other harms, and Complications/adverse events in total (also listed in Appendix A in S1 File).

## Objectives

In this study, we aimed to answer the following research questions (RQs):

1) How many publications from RCTs and SRs report physical harms related to CRCSPs?

2) How many publications from RCTs and SRs have had prior published protocols and were there any deviations concerning harm reporting between protocols and publications?

3) On the study level, how many of the 17 types of physical harm were reported (study coverage)?

4) On an outcome level, how many studies investigated each type of harm (outcome coverage)?

5) What degree of harm severity was reported in the RCTs and SRs for each type of harm?

Methods pertaining to each of the research questions above are outlined following the overall methodological considerations for the overview below.

## Methods addressing the overview of SRs and RCTs

**Identification of RCTs and SRs.** For this overview, we identified relevant RCTs from the reference review, which was conducted from 2017 to 2024 following the publication of a review protocol on PROSPERO [18]. The reference review's search strategy was conducted in multiple databases (PubMed, Embase, CINAHL, PsycINFO, and the Cochrane Library) and is available in Appendix B in S1 File.

To identify SRs, we restricted the systematic search strategy from the reference review to SRs in the PubMed database. In addition, we included SRs known to the author group, and we scrutinised the reference list of all included RCTs and SRs to find additional SRs not identified via the search. The search string was last updated on July 10th, 2024. Our definition of systematic reviews followed the NIH definition, defining a systematic review as *"a review of primary literature in health and health policy that attempts to identify, appraise, and synthesize all the empirical evidence that meets specified eligibility criteria to answer a given research question. Its conduct uses explicit methods aimed at minimizing bias to produce more reliable findings regarding the effects of interventions for prevention, treatment, and rehabilitation that can be used to inform decision making* [19]*"*.

**Eligibility assessment of RCTs and SRTs.** We assessed the eligibility of RCTs and SRs using a standard PRISMA sorting process on title, abstract and then full text level by two reviewers independently, involving a third review author when needed. Appendix C in S1 File lists the eligibility criteria. Of note, we included RCTs no matter the number of participants and the risk of bias.

## Management of overlap between study populations in systematic reviews and RCTs

The reference review revealed that reporting across multiple publications from the same RCTs was insufficient to clearly identify the extent of overlap between study populations when findings were presented in separate publications from overlapping screening rounds. This overview aimed to evaluate the number and adequacy of RCT and SR publications reporting physical harms from CRCSPs. Therefore, assessing overlap between study populations in these publications was not a concern. When we found multiple publications from the same RCT, we chose the most recent publication and extracted data concerning reporting of harms from this publication, if all data from the first publication(s) were included in the final publication. If not, both studies were included as separate studies. For SRs reporting their results in more than

one publication, we combined these data. Of note, when we from here on refer to "study" coverage and "studies", it is synonymous with the most recent publication from the study.

**Data extraction.** For RCTs and SRs, we extracted data about harm reporting, including the type of harm reported, how harms were defined, and if the consequences of the type of harm were described. We extracted information about all types of physical harm regarding severity, potential causality, and consequences. Harms had to occur during the diagnostic work-up phase of CRCSPs, i.e., due to sigmoidoscopy and/or colonoscopy, to be included. Appendix D in S1 File lists all variables extracted from RCTs and SRs with rules for assumptions made and/or measures taken to identify and clarify missing or unclear information.

**Assessment of the internal and external validity of findings from included studies.** We did not assess the risk of bias (internal validity) nor the certainty of the evidence, e.g., using GRADE (external validity), as the aim of the study was to assess the quality of reporting and not the validity of study findings. The reference review provides risk of bias assessments and GRADE ratings of all RCTs included in this overview [20].

## Methods addressing the objectives of the overview

**RQ1) Publications from RCTs and SRs reporting physical harms of CRCSPs.** Following the methods described above concerning search strategy, eligibility assessment and considerations about overlap between study populations, we counted the number of the most recent publications from RCTs and SRs reporting physical harms of CRCSPs.

**RQ2) Harm reporting in protocols compared to final publications.** We searched for protocols for studies in published articles and via databases for clinical studies, e.g., ClinicalTrials.gov and PROSPERO. We scrutinised available protocols and extracted information about intended harm assessment and reporting.

Subsequently, we compared the actual reporting of physical harms in the publications with the intended reporting in the studies' corresponding protocols and noted deviations. In case a study protocol stated that a given harm would be investigated, e.g., perforation, and this harm was not reported in the final publication, it was noted as a deviation. It was also noted as a deviation when the study protocol did not specify how harm would be reported in the final publication. If a study mentioned a protocol without referring to it, it was also noted.

**RQ3) Study coverage.** Using the 17 types of physical harm as a reference list (Appendix A in S1 File), we assessed study coverage for each study, i.e., how many of the 17 types of physical harm each RCT and SR reported.

**RQ4) Outcome coverage.** Still using the 17 types of physical harm as a reference list, we examined outcome coverage by looking at how many of the studies (RCTs and SRs, respectively) reported on each specific type of harm.

**RQ5) Severity assessment.** We tried to identify an existing classification system of complications, but found none that were appropriate to use on complications from CRCSPs. For that reason, we developed a severity assessment: for all types of harm, we distinguished between five potential severity categories: Not Severe, Severe, Very Severe, Death, and Unknown. We then categorised each type of harm reported in RCTs and SRs according to these five categories. We used information from studies to categorise the severity of the type of harm. When an outcome had a fatal outcome, it was categorised as "Death". When harms were vaguely defined or when the consequences of the type of harm were not reported, we categorised the harm assessment as "Unknown". We distinguished between "Severe" and "Very severe", categorising outcomes as the latter when they led to further surgical procedures or prolonged hospitalisation. The fifth category, "Not severe", was used when authors described the outcome as not severe or specified that the outcome led to a minimum of discomfort. The developed rules of categorisation can be seen in Appendix E in S1 File.

Subsequently, we created an overview of how often each of the five potential severities of each of the 17 types of physical harm were reported in RCTs and SRs. Some studies reported different severities of harm, e.g., mild and severe bleeding, contributing data both for "Severe" and "Not severe" bleeding. Of note, when counting the number of severity assessments across studies, we did not account for the number of harms occurring in each study, e.g., one study

reporting seven severe bleedings and 15 bleedings that were not severe counted as two severity assessments. In other words, this research question deals with the consequences of an outcome and its level of severity at study level.

## Results

### 1) How many RCTs and SRs investigate the potential physical harms of CRCSPs?

**Randomised controlled trials about the physical harms of CRCSPs.** The search strategy in the reference review identified 17058 publications after duplicates were removed. Following an eligibility assessment on title, abstract, and full-text level, 69 studies were included. An update of the search strategy in 2022 (and final update in 2023) added 35 studies, and scrutinising the reference lists of studies included for review added 30 records, leaving 134 studies to be included in the reference review. Of these, 27 were RCTs (please view Appendix F(a) in S1 File). Three of these RCTs were excluded from this study; Two due to being feasibility studies, and one because it was ongoing and not yet published in full-text format. Thus, 24 RCTs (Table 1) were included in the current study [21–44]. To view a detailed illustration of the review process, see Fig 2.

**Systematic reviews about the physical harms of CRCSPs.** The search for relevant SRs in PubMed identified 187 publications. Following screening on title and abstract level, 18 publications were read at full-text level (please view Appendix F(b) in S1 File). Here, eight studies were excluded (Appendix F in S1 File), and six publications were added after reviewing the reference lists. Thus, 16 SRs (Table 2) were included in the present study [9,17,45–58] (Fig 3).

### 2) Harm reporting in protocols compared to final publications

**Randomised controlled trials.** 14 of the 24 RCTs (58%) had published protocols. Of these 14 RCTs, 7 (50%) had deviations between the reporting of harms in the protocol compared to the reporting in the final publication (Table 1). In two studies, we found deviations from the protocols that were not reported by the authors: Bretthauer et al. [31] had a protocol [59] which stated that the authors would measure pain and discomfort during and after the screening examination. However, discomfort was *not* reported in the published article [31], whereas pain was.

Atkin et al. [39] had a protocol [60] stating that "*all adverse events which are possibly related to any aspect of the screening procedure*" were documented, and that the authors would measure these outcomes at 3-, 6- and 12 months. However, the published publication only reported adverse events after 3 months, but not after 6 and 12 months.

The protocols of Mandel et al., Kewenter et al., Hoff et al., Kobiela et al., and Holme et al. did not specify whether harms would be reported, and if so, how, yet ended up doing so in the published RCTs [28,34,35,41,42]. Both studies by Hol et al. [37,38] referred to a protocol in the published articles without any reference, and we were not able to find it by deliberately searching the abovementioned databases. Therefore, it is registered as not having a protocol in this study.

**Systematic reviews.** Of the 16 SRs included in this study, seven (44%) had published protocols [9,17,47,50,56,57,61]. Of these, three were found to have deviations between the reporting of harms in the protocol compared to reporting in the final publication (Table 2).

Huffstetler et al. had a protocol [62] stating that included studies were "*good-quality population based studies of the following study types: Systematic reviews, Randomized, controlled trials, Selected well-designed controlled clinical trials, Cohort studies. Studies might be prospective or retrospective depending on their design*". In the published article it is merely stated that included studies are "*prospective or retrospective analysis of the harms of colonoscopy*". Furthermore, it was stated in the protocol, that the outcomes examined were "*harms to patients, defined by gastrointestinal perforation, gastrointestinal bleeding, infection, or mortality directly occurring from a screening colonoscopy procedure*", however in the published article specified that "*serious adverse events other than bleeding or perforation were not included in their analysis after extraction due to inconsistent reporting and methods of data collection*".

**Table 1. Included RCTs and a summary of the corresponding protocols.**

| Study ID | Country | Study period | People | Sex, women | Age | Protocols | Deviation from protocol |
|---|---|---|---|---|---|---|---|
| Atkin 2002 [39] | UK | 1996-1999 | 42805 | 49.55% | 55-64 years | Yes | Yes |
| Bretthauer 2016 [31] | Poland, Norway, the Netherlands, and Sweden | 2009-2014 | 11912 | 48.4% | 55-64 years | Yes | Yes |
| Forbes 2006 [26] | Australia | 2004 | 151 | Not Reported | 50-69 years | No | Not relevant (NR) |
| Fritzell 2020 [43] | Sweden | 2014-2016 | 6524 | 49.0% | 59-60 years | Yes | No |
| Gondal 2003 [29] | Norway | 1999-2000 | 15484 | 51.4% | 50-64 years | Yes | No |
| Hoff 2009 [35] | Norway | Jan 1999 and Dec 2000 | 8846 | 51.48% | 55-64 years | Yes | Yes |
| Hol 2010a [37] | Netherlands | 2006-2007 | 1854 | Not reported | 50-74 years | No | NR |
| Hol 2010b [38] | Netherlands | Nov 2006 – May 2008 | 2264 | gFOBT: 54.7%, FIT: 49.4%, FS: 49.3% | 50-74 years | No | NR |
| Holme 2014 [42] | Norway | 1999-2011 | 15701 | 50.1% | 50-64 years | Yes | Yes |
| Kewenter 1996 [28] | Sweden | 1990 | 2293 | Not reported | 60-64 years | Yes | Yes |
| Kobiela 2020 [41] | Poland | 2012-2015 | 55390 | 53.5% | 55-64 years | Yes | Yes |
| Larsen 2002 [27] | Norway | Jan 1999 and Feb 2000 | 4956 | Not reported | 55-64 years | No | NR |
| Mandel 1993 [34] | US | 1975-1992 | 11943 | Not reported | 50-80 years | Yes | Yes |
| Quintero 2012 [24] | Spain | 2009-2011 | 5722 | 53.4% | 50-69 years | Yes | No |
| Randel 2021 [44] | Norway | 2012-2019 | 46307 | 37.4% − 50.8% | 50-74 years | Yes | No |
| Rasmussen 1999 [32] | Denmark | 1992-1995 | 2725 | Not reported | 50-75 years | No | NR |
| Robinson 1999 [36] | UK | 1981-1995 | 1246 | Not reported | 45-74 years | No | NR |
| Schoen 2012 [25] | US | 1993-2001 | 84305 | Not reported | 55-74 years | No | NR |
| Segnan 2002 [22] | Italy | 1995-1999 | 10686 | 46.8% | 55-64 years | No | NR |
| Segnan 2005 [33] | Italy | 1999-2001 | 4864 | 50.0% | 55-64 years | No | NR |
| Senore 2011 [21] | Italy | 2002-2004 | 3078 | 49.1% − 49.3% | 55-64 years | Yes | No |
| Stoop 2012 [30] | Netherlands | 2009-2010 | 1276 | 49.0% | 50-74 years | Yes | No |
| Van Dam 2013 [40] | Netherlands | 2010 | 322 | 48.0% | 50-74 years | Yes | No |
| Wijkerslooth 2012 [23] | Netherlands | Jun 2009 and Aug 2010 | 1276 | 50.0% | 50-74 years | Yes | No |

Table 1: The demographic characteristics of the included randomised controlled trials (in alphabetical order), including an overview of whether the studies have protocols and whether they deviate from them.

Kindt et al. and Martiny et al. shared the same protocol [18] in which it was stated that they wanted *"to report the types of physical harms of colorectal cancer screening including the risk, magnitude and consequences of these harms."* However, Martiny et al. reported only on cardiovascular complications and death. In the published article, it is stated that *"due to an unexpectedly large heterogeneity of the evidence, we chose to divide the reporting of the review's findings into separate publications. This publication reports the (…) findings related to the second to seventh objectives for the two most severe types of physical harm associated with CRCSPs, i.e., death and cardiopulmonary events (CPEs)".* Kindt et al. similarly only reported on bleedings and perforations, as they were the most frequently reported adverse events, but did not specify any further in the published article.

Reumkens et al. stated that they"(…) *applied a local protocol for conducting a meta-analysis as described elsewhere(…)*"; however, no protocol could be found.

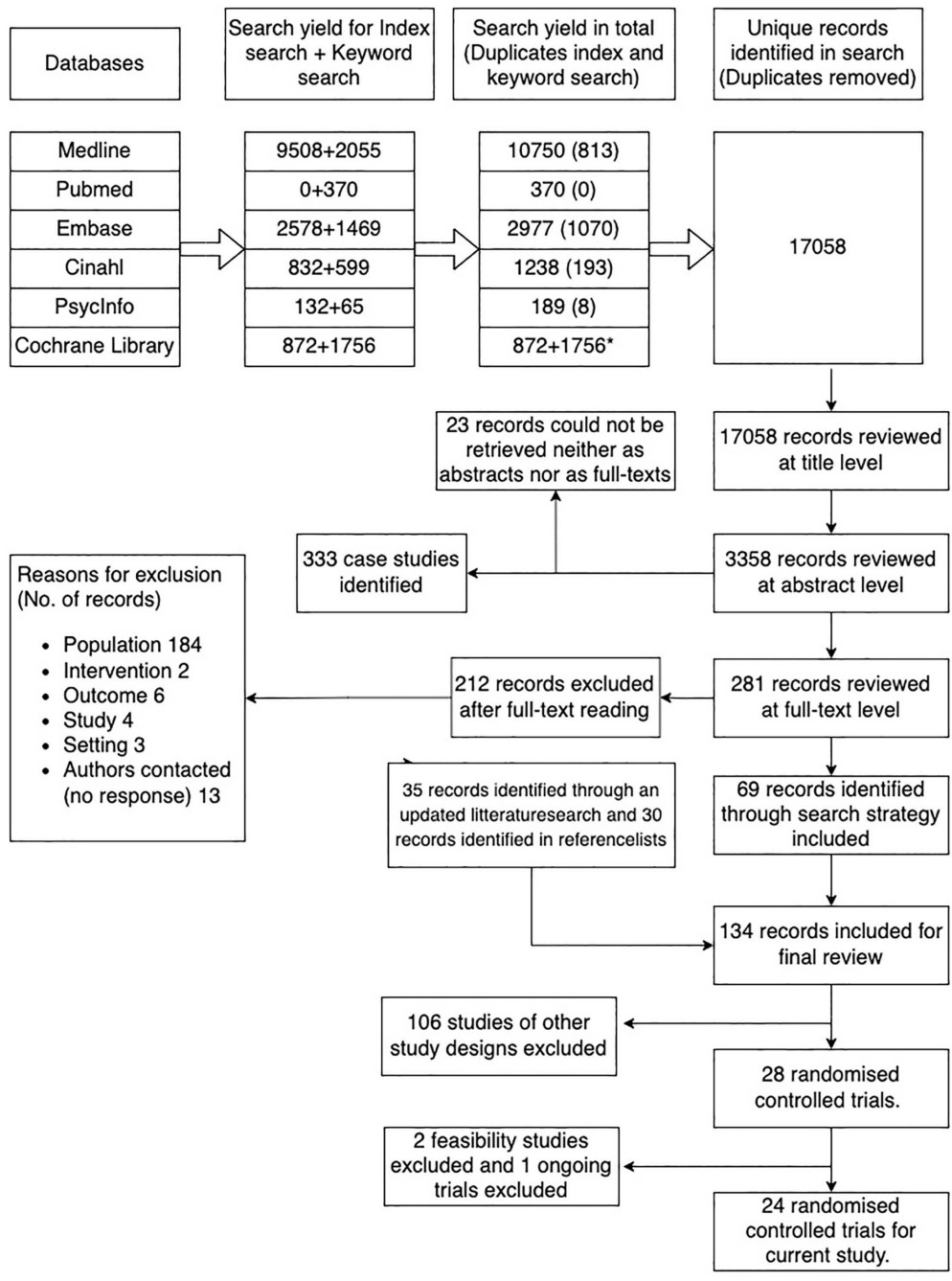

**Fig 2. RCTs.** Flowchart of the study selection process of included RCTs.

No protocols were identified for the two SRs published by Lin et al. [51,55], however, we found that it is not standard procedure for systematic reviews from the U.S. Preventive Services Task Force to publish protocols before performing a systematic review.

Table 2. Included SRs and a summary of the corresponding protocols.

| Study ID | Data sources | Time | Included study designs | Language | No. of included studies | Population | Protocols | Deviation from protocol |
|---|---|---|---|---|---|---|---|---|
| Chandan 2022 [54] | Ovid EBM Reviews, Clinical-Trials.gov, Ovid EMBASE, Ovid MEDLINE, Scopus, Web of Science. A manual search of Google Scholar | A systematic and detailed search was run in January 2021 | Included studies were observational in design, Randomised trials, Prospective controlled studies, prospective cohort, case-control, population-based, retrospective, | The results were limited to English language only | 31 records | We only included studies that reported on outcomes of colonoscopy performed in patients with positive FIT or gFOBT. (Exclusion of studies performed in the paediatric population (age,18 years), | No | NR |
| Fitzpatrick-Lewis 2016 [50] | Medline, Cochrane Library, and Embase databases. Reference lists of recent on-topic systematic reviews were checked. A targeted search of PubMed was conducted for on-topic randomised controlled trials (RCTs). | Entries dated January 2000 to February 3, 2015. (Targeted search of PubMed from February to October 2015) | "Nine RCTs addressed the benefits of screening, and 46 studies provided data for harms of screening or follow-up testing" | The search was limited to English- and French-language articles | 87 (Benefits: 9 Test properties: 37 Adverse events: 46) | Average risk, asymptomatic adults aged 18 years or older | Yes | No |
| Hewitson 2011 [49] | COCHRANE LIBRARY, MEDLINE, EMBASE, CINAHL, PSYCHINFO, AMED, SIGLE, and HMIC electronic databases | The latest update was performed in 2010 | RCTs | There were no restrictions on language of the articles | 12 records | Adults (18 years or over) participating in controlled colorectal cancer screening trials, either ongoing or completed. | No | NR |
| Holme 2013 [47] | The Cochrane Central Register of Controlled Trials (CENTRAL) (2012, Issue 11), MEDLINE and EMBASE | Until November 2012 | RCTs | There was no restriction regarding publication language | 22 records | Adult (18 years and older) asymptomatic individuals participating in a CRC screening trial with either FOBT or flexible sigmoidoscopy. | Yes | No |
| Huffstetler 2023 | Eligible trials for harms were identified by searches of PubMed and Embase | Studies published between January 1, 2002 and April 1, 2022 | Prospective or retrospective analysis of the harms of colonoscopy | English-language studies | 6 records | NR (... "excluded patients for whom colonoscopy was likely for diagnostic purposes rather than screening, for example, surveillance colonoscopies, high risk patients, patients with symptoms of colorectal cancer, or those with a personal history of inflammatory bowel disease or colorectal cancer" | Yes | Yes |

(Continued)

**Table 2.** (Continued)

| Study ID | Data sources | Time | Included study designs | Language | No. of included studies | Population | Protocols | Deviation from protocol |
|---|---|---|---|---|---|---|---|---|
| Jodal 2019 [53] | Update of a previously performed Cochrane review search Database searching References | The search previously ended in 2012 and was updated until December 2018 | RCTs | No language restrictions | Articles included in review 36 (RCTs included in review 12) | A healthy population aged 50–79 | Yes | No |
| Kayal 2023 | MEDLINE, EMBASE and PsycINFO | The search was limited to studies published between 2000 and 2021 | Retrospective, prospective and cross-sectional survey studies, including at least one patient-reported outcome | English-languaged studies | 6 records | Colorectal cancer screening participants, age ranged from 50 to 75 years old. | Yes | No |
| Kindt 2023 | PubMed, MEDLINE, Embase, CINAHL, PsycINFO and the Cochrane Library | The search was conducted on April 12th 2017 and updated March 22th 2022 | Studies were included, regardless of study design | No language restrictions | 89 records | People at average risk of colorectal cancer, i.e., a general screening population. | Yes | No |
| Lin 2016 [51] | MEDLINE, PubMed, the Cochrane Central Register of controlled trials. Citations and references The search also included selected gray literature sources, including Clinical-Trials.gov and the World Health Organization International Clinical Trials Registry Platform, for ongoing trials. | January 2008-December 2014 (Surveillance until February 2016) | KQ1: RCTs, otherwise controlled trials, well-conducted prospective cohort or population-based nested case-control studies. KQ2: Diagnostic accuracy studies (exclusion of case-controlled studies) KQ3: All trials and observational studies that reported serious adverse events | English language studies | In total: 204 (KQ1: 47, KQ2: 44, KQ3: 113) | Asymptomatic screening population of 40 years or older, at average risk of CRC | No | NR |
| Lin 2021 [55] | MEDLINE, PubMed, and the Cochrane Central Register of Controlled Trials Expert suggestions and references | Studies published from January 1, 2015, to December 4, 2019; surveillance through March 26, 2021. | KQ1: RCTs or nonrandomised controlled intervention studies For KQ2, test accuracy studies, well-conducted test accuracy studies For KQ3, all trials and observational studies | English-language studies | In total: 306 articles, 223 studies. (KQ1: 66 articles, 33 studies, KQ2: 78 articles, 59 studies, KQ3: 162 articles, 131 studies) | Asymptomatic populations, of individuals 40 years or older, at general risk of CRC. | No | NR |

*(Continued)*

| Study ID | Data sources | Time | Included study designs | Language | No. of included studies | Population | Protocols | Deviation from protocol |
|---|---|---|---|---|---|---|---|---|
| Martiny 2024 | MEDLINE (via PubMed), Embase, CINAHL, PsycINFO and the Cochrane Library | All databases were searched from their inception date to the 4th of March 2022 | No restrictions regarding study design | No restrictions regarding language | 134 records | People at average risk of colorectal cancer, i.e., a general screening population. | Yes | Yes |
| Niv 2008 [48] | PubMed (Medline), EMBASE, Cochrane Library (Cochrane Database of Systemic Reviews and the Cochrane Controlled Trial Register), CINAHL, and AMED databases References | Up to October 2007 | Cohort studies | Papers appearing in English-language medical journals. | 10 records | Asymptomatic people, defined as men and women aged 40–75 years, without symptoms of abdominal pain, rectal bleeding or a change in bowel habits. | No | NR |
| Reumkens 2016 | Pubmed, the Cochrane library, and Ovid EMBASE | Inclusion of population-based studies examining post-colonoscopy complications in patients undergoing colonoscopy from January 2001 until 01 December 2015. | Retrospective studies, prospective studies | Studies published in English language were included. | 21 records | Mean age of the study populations varied from 50.4 to 70.0 years (4,6,10,14–21,24,25) (median age, 57–75 years (5,9,13,16,21)); 51.5% were males (4–9,11–22,24,25). | No | NR |
| Tinmouth 2016 [45] | OVID MEDLINE, EMBASE, the Cochrane Library and the American Society of Clinical Oncology (ASCO) conference proceedings | OVID MEDLINE (2006 to September 3, 2014), EMBASE (2006 to September 3, 2014), the Cochrane Library (Issue: 2–4, October 2013), and ASCO conference proceedings (2009–2013) | Systematic reviews, RCTs, prospective studies, retrospective studies, case-control studies | English-language publications | 76 records | Asymptomatic average risk subjects undergoing CRC screening | No | NR |
| Towler 1998 [52] | An update of a review published by some of the authors in 1995. References MEDLINE, Cochrane Controlled Trials Register | MEDLINE: 1990–1996 Current contents fra January 1996-January 1997. Cochrane Controlled Trials Register: January 1997 | Randomised and non randomised controlled trials | NR | 6 records | Adults participating in the Hemoccult screening trials being conducted around the world. The age of participants varies from trial to trial with all being at least 40 years old. | No | NR |
| Vermeer 2017 [46] | PubMed, MEDLINE, Embase, Web of science and PsychINFO (EBSCO) References | Each databases' inception to August 2016 | All types of studies, both retrospective and prospective | Publications in English language | 60 records | Adults (exclusion when a study reported outcome of patient with increased risks). No age specified | No | NR |

Table 2: The demographic characteristics of the included systematic reviews (in alphabetical order), including an overview of whether the studies have protocols and whether they deviate from them.

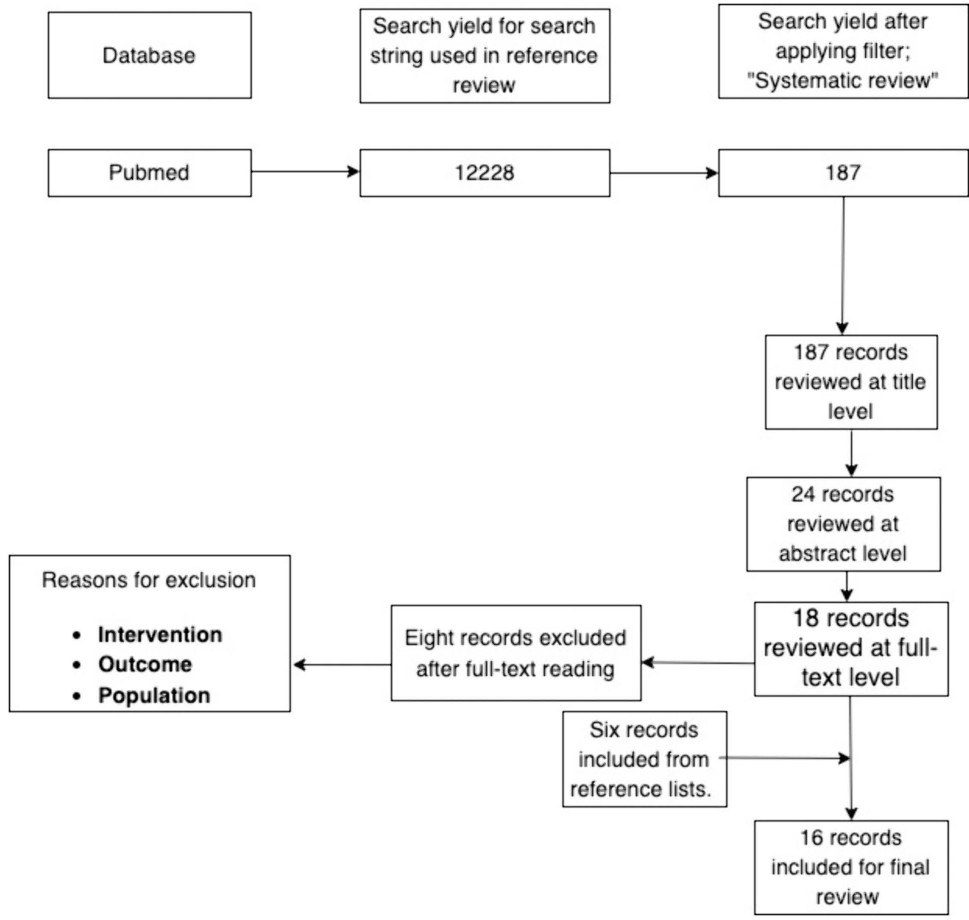

**Fig 3. SRs.** Flowchart of the study selection process of included SRs.

### 3) Study coverage

**Randomised controlled trials.** Across the 24 RCTs, 16 of the 17 (94%) types of physical harm were reported at least once (Table 3).

The average number of physical harms reported in RCTs was 4.1/17 (24%). The median number of types of harm reported was also 4/17 (24%). The greatest study coverage was 8/17 (47%) in four studies [21,22,30,31]. The lowest study coverage was 1/17 (6%) in three RCTs [25,32,35] (Table 3). A combined overview of study coverage, as well as outcome coverage for the RCTs, is available in Table 5a in Appendix G in S1 File.

**Systematic reviews.** Across the 16 SRs, 14/17 (82%) types of physical harm were reported at least once (Table 4).

The average number of types of harm reported in the SRs was 4.8/17 (28%). The median number of reported types of harm was 3/17 (18%).

The greatest study coverage was 9/17 (53%) in Lin et al. [51], while the lowest was 1/17 (6%) in Tinmouth et al. [45]. A combined overview of study coverage and outcome coverage for the SRs is available in Table 5b in Appendix I in S1 File.

### 4) Outcome coverage

**Randomised controlled trials.** Outcome coverage in RCTs varied from no RCTs reporting a given type of harm (0%), to 16/24 (67%) (Table 5). The type of harm with the highest outcome coverage was "Bleeding", reported in 16 of 24 RCTs (67%). None of the RCTs reported on Sedation-related complications.

**Table 3. Study coverage in RCTs.**

| Study ID, RCTs: | Study coverage: | Types of harm*: |
|---|---|---|
| Senore 2011 | 8/17 (47.1%) | Cardiovascular- and pulmonary complications, Bleeding, Colorectal symptoms, Complications related to bowel preparation, Dizziness, Pain, Other harms, and Complications/adverse events in total |
| Segnan 2002 | 8/17 (47.1%) | Perforation, Cardiovascular- and pulmonary complications, Bleeding, Inflammatory complications, Pain, Discomfort, Other harms**, and Complications/adverse events in total |
| Bretthauer 2016 | 8/17 (47.1%) | Death, Perforation, Cardiovascular- and pulmonary complications, Bleeding, Post-polypectomy syndrome, Pain, Other harms and Complications/adverse events in total |
| Stoop 2012 | 8/17 (47.1%) | Death, Cardiovascular- and pulmonary complications, Bleeding, Infections, Inflammatory complications, Pain, Other harms and Complications/adverse events in total |
| Hol 2010b | 7/17 (41.2%) | Bleeding, Colorectal symptoms, Complications related to bowel preparation, Nausea/vomiting, Pain, Discomfort, and other harms |
| Atkin 2002 | 6/17 (35.3%) | Death, Perforation, Cardiovascular- and pulmonary complications, Bleeding, Inflammatory complications, and Pain |
| Gondal 2003 | 5/17 (29.4%) | Perforation, Bleeding, Post-polypectomy syndrome, Complications related to bowel preparation, and Other harms |
| Robinson 1999 | 5/17 (29.4%) | Death, Perforation, Bleeding, Other harms, and Complications/adverse events in total |
| Wijkerslooth 2012 | 5/17 (29.4%) | Colorectal symptoms, Complications related to bowel preparation, Sleep disturbances, Pain, and Other harms |
| Forbes 2006 | 4/17 (23.5%) | Perforation, Bleeding, Pain, and Complications/adverse events in total |
| Quintero 2012 | 4/17 (23.5%) | Perforation, Cardiovascular- and pulmonary complications, Bleeding, and Complications/adverse events in total |
| Van Dam 2013 | 4/17 (23.5%) | Cardiovascular- and pulmonary complications, Colorectal symptoms, Nausea/vomiting, and Other harms |
| Holme 2014 | 4/17 (23.5%) | Death, Perforation, Bleeding and Complications/adverse events in total |
| Segnan 2005 | 3/17 (17.6%) | Cardiovascular- and pulmonary complications, Bleeding, and Pain |
| Fritzell 2020 | 3/17 (17.6%) | Pain, Discomfort and Complications/adverse events in total |
| Randel 2021 | 3/17 (17.6%) | Death, Perforation and Bleeding |
| Kewenter 1996 | 3/17 (17.6%) | Bleeding, Perforation and Complications/adverse events in total |
| Hol 2010a | 2/17 (11.8%) | Bleeding and Other harms |
| Larsen 2009 | 2/17 (11.8%) | Pain and Discomfort |
| Mandel 1993 | 2/17 (11.8%) | Perforation and Bleeding |
| Kobiela 2019 | 2/17 (11.8%) | Death and Other harms |
| Hoff 2009 | 1/17 (5.9%) | Complications/adverse events in total |
| Rasmussen 1999 | 1/17 (5.9%) | Complications/adverse events in total |
| Schoen 2012 | 1/17 (5.9%) | Perforation |

*All 17 types of harms: Death, Perforation, Cardiovascular- and pulmonary complications, Bleeding, Post-polypectomy syndrome, Infections, Inflammatory complications, Colorectal symptoms, Sedation-related complications, Complications related to bowel preparation, Sleep disturbances, Nausea/vomiting, Dizziness, Pain, Discomfort, Other harms*, and Complications/adverse events in total.

**Other harms comprise many different types of harm. Please view Appendix F in S1 File.

The average outcome coverage for RCTs was 5.8/24 (24%), i.e., on average, each type of harm was reported in roughly one in four RCTs. The median outcome coverage was 4/24 (17%).

**Systematic reviews.** Outcome coverage in SRs varied from 0/16 (0%) to 14/16 (88%) (Table 6). The types of harm most frequently reported were "Perforation" and "Bleeding," both reported in 14 of 16 reviews (88%). The average outcome coverage was 4.5/16 (28%), i.e., on average, every type of harm was reported on in approximately a third of SRs. The median outcome coverage was 3/16 (19%).

None of the SRs reported on Sleep disturbances, Nausea/vomiting, or Dizziness.

## 5) Severity of physical harms reported

Tables 7 and 8 present the number of RCTs and SRs, respectively, reporting on the five categories of severity for each of the 17 types of physical harm.

**Table 4. Study coverage in SRs.**

| Study ID, SRs: | Study coverage: | Types of harm*: |
|---|---|---|
| **Lin 2016** | 9/17 (52.9%) | Death, Perforation, Cardiovascular- and pulmonary complications, Bleeding, Infections, Inflammatory complications, Colorectal symptoms, Pain, and Other harms |
| **Jodal 2019** | 8/17 (47.1%) | Death, Perforation, Cardiovascular- and pulmonary complications, Bleeding, Post-polypectomy syndrome, Pain, Other harms and Complications/adverse events in total |
| **Vermeer 2017** | 8/17 (47.1%) | Death, Perforation, Cardiovascular- and pulmonary complications, Bleeding, Post-polypectomy syndrome, Pain, Discomfort, and Complications/Adverse events in total |
| **Chandan 2011** | 8/17 (47.1%) | Death, Perforation, Cardiovascular – and pulmonary complications, Bleeding, Post-polypectomy syndrome, Sedation-related complications, Other harms and Complications/Adverse events in total |
| **Kayal 2023** | 8/17 (47.1%) | Perforation, Cardiovascular – and pulmonary complications, Bleeding, Post-polypectomy syndrome, Pain, Discomfort, Complications related to bowel preparation and other harms |
| **Holme 2013** | 7/17 (41.2%) | Death, Perforation, Cardiovascular – and pulmonary complications, Bleeding, Inflammatory complications, Other harms, and Complications/adverse events in total |
| **Hewitson 2011** | 6/17 (35.3%) | Death, Perforation, Bleeding, Inflammatory complications, Other harms and Complications/adverse events in total |
| **Fitzpatrick-Lewis 2016** | 3/17 (17.6%) | Death, Perforation, and Bleeding, |
| **Niv 2008** | 3/17 (17.6%) | Death, Perforation, and Bleeding |
| **Lin 2021** | 3/17 (17.6%) | Perforation, Bleeding and Other harms |
| **Huffstetler 2023** | 3/17 (17.6%) | Perforation, Cardiovascular – and pulmonary complications and Bleeding |
| **Reumkens 2016** | 3/17 (17.6%) | Death, Perforation and Bleeding |
| **Kindt 2023** | 3/17 (17.6%) | Death, Perforation and Bleeding |
| **Martiny 2024** | 2/17 (11.8%) | Death and Cardiovascular- and pulmonary complications |
| **Towler 1998** | 2/17 (11.8%) | Perforation and Bleeding |
| **Tinmouth 2016** | 1/17 (5.9%) | Complications/adverse events in total |

*All 17 types of harms: Death, Perforation, Cardiovascular- and pulmonary complications, Bleeding, Post-polypectomy syndrome, Infections, Inflammatory complications, Colorectal symptoms, Sedation-related complications, Complications related to bowel preparation, Sleep disturbances, Nausea/vomiting, Dizziness, Pain, Discomfort, Other harms*, and Complications/adverse events in total.

**Other harms comprise many different types of harm. Please view Appendix H in S1 File.

Many types of harm lacked sufficient information to categorise the type of harm according to its severity, leading us to categorise them as Unknown.

**Randomised controlled trials.** Across the 24 RCTs, 16/17 (94%) types of physical harm were reported. We assessed the severity of 15 types of harm since death has no meaningful severity assessment and no studies reported Sedation-related complications. Non-severe occurrences were reported for 9/15 (60%) types of harm. Overall, 10/15 (67%) types of harm were assessed as Severe. In 5/15 (33%) types of physical harm outcomes were defined as being Very severe. Of note, 8 of the 15 (53%) types of harm reported in RCTs were classified as Unknown severity due to vague definitions or incomplete reporting of the consequences of harm, making it impossible to assess the level of severity (Table 7).

As an example, Table 7 illustrates that 7/22 (31.8%) of the assessments of Bleeding were classified as "Unknown" due to incomplete reporting, impeding distinct severity classification.

Across the 16 types of harm assessed, 173 different harm severities were reported in RCTs (Table 6, Appendix J in S2 File) and 30 (17.3%) of these were sorted in the Unknown category due to incomplete reporting.

## Systematic reviews

Table 8 shows the severity assessment of the 14/16 (88%) types of harm reported in the 16 SRs.

**Table 5. Outcome coverage in RCTs.**

| Outcomes**: | Outcome coverage |
|---|---|
| Bleeding | 16/24 (66.7%) |
| Perforation | 12/24 (50%) |
| Pain | 12/24 (50%) |
| Complications/adverse events in total | 12/24 (50%) |
| Other harms* | 11/24 (45.8%) |
| Cardiovascular-and pulmonary complications | 8/24 (33.3%) |
| Death | 7/24 (29.2%) |
| Colorectal symptoms | 4/24 (16.7%) |
| Complications related to bowel preparation | 4/24 (16.7%) |
| Discomfort | 4/24 (16.7%) |
| Inflammatory complications | 3/24 (12.5%) |
| Nausea/vomiting | 2/24 (8.3%) |
| Post-polypectomy syndrome | 2/24 (8.3%) |
| Infections | 1/24 (4.2%) |
| Dizziness | 1/24 (4.2%) |
| Sleep disturbances | 1/24 (4.2%) |
| Sedation-related complications | 0/24 (0%) |

*Other harms comprise many different types of harm. Please view Appendix H in S1 File.

** Studies reporting on any type/severity of the outcome were included.

**Table 6. Outcome coverage in SR.**

| Outcomes**: | Outcome coverage |
|---|---|
| Bleeding | 14/16 (87.5%) |
| Perforation | 14/16 (87.5%) |
| Death | 10/16 (62.5%) |
| Cardiovascular- and pulmonary complications | 8/16 (50%) |
| Other harms* | 7/16 (43.8%) |
| Complications/adverse events in total | 6/16 (37.5%) |
| Pain | 4/16 (25%) |
| Post-polypectomy syndrome | 4/16 (25%) |
| Inflammatory complications | 3/16 (18.8%) |
| Discomfort | 2/16 (12.5%) |
| Infections | 1/16 (6.25%) |
| Colorectal symptoms | 1/16 (6.25%) |
| Sedation-related complications | 1/16 (6.25%) |
| Complications related to bowel preparation | 1/16 (6.25%) |
| Nausea/vomiting | 0/16 (0%) |
| Dizziness | 0/16 (0%) |
| Sleep disturbances | 0/16 (0%) |

*Other harms comprise many different types of harm. Please view Appendix H in S1 File.

** Studies reporting on any type/severity of the outcome were included.

**Table 7. Severity assessment.**

| Outcomes//Levels of severity | Severity assessments across included RCTs** (N) | Not severe | Severe | Very severe | Unknown | Death |
|---|---|---|---|---|---|---|
| **Bleeding** | 22 | 3 (13.6%) | 7 (31.8%) | 5 (22.7) | 7 (31.8%) | – |
| **Pain** | 20 | 9 (45%) | 11 (55%) | 0 | 0 | – |
| **Other harms** | 17 | 5 (29.4%) | 5 (29.4%) | 1 (5.9%) | 6 (35.3%) | – |
| **Perforation** | 15 | 0 | 8 (53.3) | 7 (46.7%) | 0 | – |
| **Complications/adverse events in total:** | 14 | 0 | 5 (35.7%) | 2 (14.3%) | 7 (50%) | – |
| **Cardiovascular-and pulmonary complications** | 11 | 6 (54.4%) | 2 (18.2%) | 3 (27.3%) | 0 (0%) | – |
| **Death*:** | 7 | – | – | – | – | 7 (100%) |
| **Complications related to bowel preparation** | 6 | 4 (66.7%) | 1 (16.7%) | 0 | 1 (16.7%) | – |
| **Colorectal symptoms** | 6 | 2 (33.3%) | 1 (16.7%) | 0 | 3 (50%) | – |
| **Discomfort** | 4 | 4 (100%) | 0 | 0 | 0 | – |
| **Inflammatory complications** | 4 | 0 | 3 (75%) | 0 | 1 (25%) | – |
| **Post-polypectomy syndrome** | 2 | 1 (50%) | 1 (50%) | 0 | 0 | – |
| **Nausea/vomiting** | 2 | 2 (100%) | 0 | 0 | 0 | – |
| **Infections** | 1 | 0 | 0 | 0 | 1 (100%) | – |
| **Dizziness** | 1 | 1 (100%) | 0 | 0 | 0 | – |
| **Sleep disturbances** | 1 | 0 | 0 | 0 | 1 (100%) | – |
| *Sedation-related complications* | 0 | 0 | 0 | 0 | 0 | – |

*Death could only be classified in the death severity.

** Some studies report different consequences of harms and thereby multiple severity categories of a physical harm.

Table 8. Severity Assessment of Physical Harm Reported in 24 RCTs.

**Table 8. Severity assessment.**

| Outcomes//Levels of severity | No. of severity assessments across included SRs** | Not severe | Severe | Very severe | Unknown | Death |
|---|---|---|---|---|---|---|
| **Bleeding** | 25 | 3 (12%) | 11 (44%) | 4 (16%) | 7 (28%) | – |
| **Perforation** | 17 | 0 | 13 (76.5%) | 3 (17.6%) | 1 (5.9%) | – |
| **Cardiovascular- and pulmonary complications** | 15 | 4 (26.7%) | 0 | 5 (33.3%) | 6 (40%) | – |
| **Other harms** | 10 | 2 (20%) | 2 (20%) | 0 | 6 (60%) | – |
| **Complications/adverse events in total** | 9 | 2 (22.2%) | 3 (33.3%) | 0 | 4 (44.4%) | – |
| **Death** | 10 | – | – | – | – | 10 (100%) |
| **Inflammatory complications** | 4 | 0 | 1 (25%) | 0 | 3 (75%) | – |
| **Pain** | 5 | 2 (40%) | 2 (40%) | 0 | 1 (20%) | – |
| **Post-polypectomy syndrome** | 4 | 0 | 1 (25%) | 0 | 3 (75%) | – |
| **Colorectal symptoms** | 1 | 0 | 1 (100%) | 0 | 0 | – |
| **Discomfort** | 2 | 2 (100%) | 0 | 0 | 0 | – |
| **Infections** | 1 | 0 | 0 | 0 | 1 (100%) | – |
| **Sedation-related complications** | 1 | 0 | 0 | 0 | 1 (100%) | – |
| **Complications related to bowel preparation** | 1 | 1 (100%) | 0 | 0 | 0 | – |
| *Nausea/vomiting* | 0 | 0 | 0 | 0 | 0 | – |
| *Dizziness* | 0 | 0 | 0 | 0 | 0 | – |
| *Sleep disturbances* | 0 | 0 | 0 | 0 | 0 | – |

*Death could only be classified in the death severity.

** Some studies report on multiple consequences and thereby severity categories of a physical harm.

Table 8. Severity Assessment of Physical Harm Reported in 16 SRs.

The 16 SRs reported 14/17 (88%) types of physical harm, combined. When excluding death, seven outcomes of 13 (54%) were assessed as being Not Severe. 8/13 (62%) were categorised as being Severe, and 3/13 (23%) were categorised as being Very Severe complications. 10/13 (77%) types of harm were all reported in some SRs but were classified as Unknown due to a lack of definitions of the consequences of the outcomes (Table 8).

As an example, Table 8 illustrates that 7/25 (28%) of the reports on Bleeding were *not* adequately defined to determine the downstream/resulting consequences of the bleeding and were therefore classified as being Unknown regarding the severity of the bleeding.

The SRs combined reported 139 harm outcomes (Table 7, Appendix K in S2 File), and 32 (23%) of these were sorted in the Unknown category due to incomplete reporting.

## Discussion

### Summary of main findings

In our overview, we identified 24 RCTs and 16 SRs investigating physical harm caused by CRCSPs. For RCTs, 14 (58%) had published protocols, and seven of these (50%) had deviations between the intended reporting of harms in the protocol and the actual reporting in the final publication. The main type of deviation was adding new outcome assessments in the final publication that were not pre-specified in the protocol. Seven SRs (43.8%) had published protocols, and three of these (43%) had deviations between intended and actual reporting. The main type of deviation was not reporting all the measured harms in the final publication.

At study level, RCTs and SRs on average reported less than one-third of the 17 types of physical harm identified in the reference review. On outcome level, the average reporting rate of the 17 types of harm was 24% for RCTs and 28% for SRs. However, the outcome reporting rate varied considerably between outcomes, e.g., the reporting rate of bleeding ranged from 0–67% and 0–88% in RCTs and in SRs, respectively. In addition, one outcome was not reported in any of the 24 RCTs (Sedation-related complications), and three outcomes were not reported in any of the 16 SRs (Nausea/vomiting, Dizziness, Sleep disturbances).

Our attempt to classify the severity of physical harm showed that most types of physical harm were seldom adequately defined to allow assessment of harm severity. In turn, when outcomes were defined insufficiently, poor reporting of consequences of physical harms often hindered meaningful severity assessments. For example, 8 of the 16 types of harm reported in RCTs (50%) were either so vaguely defined or lacked sufficient reporting of the consequences of the type of harm that it was not possible to assess the level of severity. In effect, 17% and 23% of outcomes reported in RCTs and SRs, respectively, were classified as unknown severity.

### Strengths

In this study, we combined recommendations from the best-available methodological guidelines to direct the conduct of our study, including the PRISMA guideline [13]; a 54-item checklist concerning reporting of methods in overviews of reviews [15]; and guidance from the Cochrane Collaboration, adapted to extracting reporting rates instead of outcomes [16]. Second, the reference review used for the comparison of reporting quality in RCTs and SRs follows the best available guidance for systematic reviews on the harms of medical interventions [9]. Third, the reference review from which this study has extracted its data has included *all* available literature in the area, without restrictions concerning study design, language, and publication date. Therefore, we believe that this study provides an up-to-date and comprehensive comparison of the reporting quality of physical harms in RCTs and SRs with all potential physical harms due to CRCSPs identified in the reference review [9]. Of note, we have not been able to identify studies similar to ours, which suggests that this is an original contribution to the scientific literature on CRCSPs and screening programmes more broadly.

## Limitations

It might be argued that because we could not identify any similar studies it is hard to assess the quality of the methodology we have used. However, we think that the developed method is strong, and the data we have extracted in this article is exactly as it is presented in the published studies.

A second potential limitation is the lack of a consensus-based framework for reporting the harms of screening in general [63,64], though some have been proposed [65], which led us to use the reference review as the best available reference standard for reporting harms in CRCSPs. In effect, our reference list comprising 17 types of physical harms associated with CRCSPs may not apply to other screening settings and might be categorised in other ways.

Thirdly, there is a possibility for misclassification bias due to categorisation of outcomes according to severity, e.g., we always categorised perforations as severe if nothing else was specified in the original article. Another potential driver of misclassification was that authors tended to describe outcomes on an overall level, which made it difficult to distinguish whether some of the outcomes reported were severe, e.g., reporting that 7 perforations occurred, where two of these were fatal, without specifying the consequences of the remaining 5 perforations.

Fourth, we were not able to identify an established method for assessing the severity of physical harms related to CRCSPs. While we acknowledge that the method we developed in this overview has limitations, we believe we have robustly assessed the reporting quality based on a systematic overview of harms and double-blinded data extraction.

A fifth limitation of this study is the potential for missing data, as we cannot rule out the possibility that some reported harms of screening were overlooked. The reporting of consequences varied considerably across studies and was often unclear or vague—for example, harms were inconsistently presented in text, tables, footnotes, or appendices, increasing the probability that some descriptions were missed

Lastly, we cannot rule out the possibility of language bias. Neither the reference review nor this overview imposed language restrictions on the primary articles. However, among the 16 included SRs, nine included English language articles, four had no language restrictions, one included studies in English and French, one only included studies from English-language journals, and one did not report language limitations (Table 2). This introduces a possible language bias, as relevant non-English studies may have been overlooked, potentially affecting the comprehensiveness of the evidence.

## Findings compared to other studies

Studies have shown that the harms of screening are underreported. Heleno et al. describe how RCTs often omit investigation of important harms of screening, while simultaneously providing a much more comprehensive account of the benefits of screening [11]. In addition, two studies have found that the physical harms of ongoing CRCSPs are much more frequent than reported rates in RCTs and SRs [12,66]. One of these studies found poor measurement quality of the physical harms of screening, reporting a surprisingly low sensitivity of 29% for any type of complication in the Danish Colorectal Cancer Screening Database [66]. Another study from France found that gastroenterologists only reported 52% of moderate and severe adverse events in relation to CRCSPs when compared to the account of patients and their general practitioners [67].

## Reporting of negative results and their importance

In this overview, the focus is specifically on the quality of reporting in RCTs and SRs and not on the potential biases that may affect the harm estimates. However, our findings show low study coverage on many types of harm, which could be interpreted as non-reporting bias.

The non-occurrence of rare events, exemplified by splenic rupture during colonoscopy, can result in non-reporting bias. This is generally considered justifiable within clinical research, given the rarity of these outcomes [68]. Non-reporting bias affects the validity of reviews and meta-analyses, which in turn can lead to misleading conclusions regarding the benefits and harms of an intervention [69]. Therefore, we would argue that reporting negative results, i.e.,

zero events, is as important as reporting positive findings [68]. Of note, we found that even major harmful outcomes, i.e., important non-rare events, were oftentimes not reported in studies, e.g., only *seven of 24* (29%) RCTs reported on the outcome Death as a complication of the CRCSPs. We perceive Death as the most severe complication of any intervention and would recommend that studies always report whether deaths and other severe types of harm have *or have not* occurred. In other words, studies should, at a minimum, report deaths, cardiopulmonary events, perforations, and bleeding, irrespective of the rarity of these outcomes or whether they occur. An example of the importance of reporting the occurrence of rare, harmful events is the case of post-polypectomy syndrome (PPS), which in some studies is referred to as "Burnt serosa syndrome". This is a type of harm that can be very severe [70], possibly leading to abdominal pain, fever, localised peritonitis, and occasionally sepsis and organ dysfunction. Additionally, the symptoms of PPS usually present within 12 hours but may be delayed up to 7 days following a colonoscopy [71]. This highlights the need for a sufficiently long follow-up period to ensure that such complications can be detected and accurately attributed to the screening procedure. However, only two RCTs (8%) and three SRs (27%) reported on PPS. When such harms are not reported, it is unclear if the studies not reporting on PPS have refrained from doing so because the complication did not occur, or because they did not have a sufficient follow-up time (unless follow-up time was reported), or if occurrences were simply not measured. A similar type of bias, comparable to non-reporting bias on the study-level, is publication bias, which occurs when the decision to publish a study is determined by the findings of the study. In our case, it is not unlikely that studies reporting more severe harms are more likely to be published, while those identifying less severe, yet still significant, harms may be disregarded or underreported. It is not possible to determine which of these explanations is the most accurate, as we cannot assess the impact of the bias. A full assessment of the risk of bias in all included articles can be found in the reference review [20].

## Implications for screening participants

We believe it is important to apply higher standards when reporting potential harms of population screening programmes than is currently the case. This need arises both from the generally low quality of harm reporting and from the considerable number of participants in CRCSPs, many of whom are otherwise healthy individuals exposed to potential harm. From an ethical standpoint, any harm caused by the screening test itself or by subsequent procedures following an abnormal result is less acceptable than harm occurring in individuals who are seriously ill and therefore more likely to benefit from diagnostic procedures or treatment [65]. Or, as Vermeer et al. [46] state, "*Since the lifetime risk of developing colorectal cancer for the population in the USA and the Netherlands is 2–5%, the majority of people undergoing screening is neither identified as having cancer nor its precursor-lesions. Every potential harm is therefore more worrisome.*"

## Implications for research

Currently, reporting of physical harms in RCTs and SRs is inconsistent and of poor quality compared to the standards set by available guidelines in the area [7,8], hindering adequate assessment of the safety of CRCSPs. Our findings concerning the poor reporting of harms due to CRCSP in SRs and RCTs both question the safety of CRCSPs and the ability of SRs and clinical guidelines to provide trustworthy accounts of physical harms due to CRCSPs.

This study aimed to assess the reporting quality of physical harms in randomised controlled trials (RCTs) and systematic reviews (SRs). Quantification of the most important types of harms—that is, deaths, cardiopulmonary events, perforation, and bleeding—was presented in two separate articles [17,20]. While it may be unrealistic to expect that all RCTs and SRs of a given intervention report on every possible type of harm, the ethical imperative to report all potential harms is paramount, especially because screening participants are generally healthy individuals.

The lack of reporting of zero events has been pointed out in the PRISMA-harms extension in its critique that SRs compound poor reporting of harms in clinical studies [7]. Furthermore, the zero-outcome coverage for one and four types

of harms in RCTs and SRs, respectively, shows that neither all the RCTs *combined* nor all the SRs *combined* uncover all potential harms of CRCSPs identified in the reference review, which also included non-randomised studies. This shows the value of including non-randomised studies (NRSs) when conducting a systematic review to identify *all* potential harms of an intervention.

The issue of poor reporting of physical harms due to CRCSPs in RCTs and SRs could be remedied with the development of a Core Outcome Set (COS) [63]. The absence of such a COS is surprising when guidance to develop a COS for screening exists [72], and many RCTs and SRs have been conducted in the field of CRCSPs for many years.

Potential measures to improve both study- and outcome coverage of future studies in this area, and medical interventions in general, include:

1. A consensus-based typology about how to define, measure, and report harms from screening, preferably a unified framework for potential types of harms associated with medical technology in general.

2. Optimal measurement methods concerning uniform outcome definitions, adequate follow-up time, trained outcome assessors, and systematic harm assessment.

3. The development of a Core Outcome Set (COS) for CRCSPs with standardised outcomes that should always be reported, even in the event of zero events (e.g., for deaths, bleedings, and perforations in the case of CRCSPs). This process involves engaging stakeholders, including patients, healthcare professionals, and expert panels, in conducting a comprehensive review with no restrictions on language or study design, followed by consensus processes, e.g., the Delphi method. This allows for the generation of a comprehensive list of outcomes, followed by the identification of core outcomes that should be consistently reported for all trials about CRCSPs.

### Implications for practice and policy

In many countries, citizens between the ages of 50 and 74 years are invited to CRCSPs, often having limited health literacy and overconfidence in the value of screening [73]. In addition, research has found that clinicians tasked with guiding people about decisions regarding their health, e.g., screening participation, tend to have the same misinterpretation as laypeople about the balance between harm and benefits [74]. Therefore, it is paramount that information materials about the benefits and harms of screening are of the highest quality to support adequate decision-making about screening. This requires better reporting of harms in RCTs, SRs, and guidelines, as information materials for the citizens are derived from these.

Policymakers need to be made aware of the fact that many studies provide poor reporting of physical harm due to CRCSPs, and they need to be attentive to this when scrutinising studies and developing guidelines based on them.

The development of guidelines could result in an increasingly balanced weighing between benefits and harms when considering whether to implement an intervention at a population level. We would recommend that editors demand that the existing and future guidelines be followed before publishing studies that assess the possible harms of interventions.

### Conclusion

Our results revealed the lack of a consensus-based method to adequately report on the harms of colorectal cancer screening.

Both study and outcome coverage of RCTs and SRs were 24% and 31%, respectively. One outcome, identified in the reference review, was not reported on in any of the RCTs. SRs compounded the poor reporting of harms in RCTs by omitting to report on four outcomes identified in the reference review.

For some types of harm, a substantial part of the reporting lacked a sufficient definition, causing difficulties in assessing the consequences of the reported harms and thus difficulties in assessing the safety of colorectal cancer screening.

We would argue there is a need for better use of the already existing guidelines for reporting harm in RCTs and SRs, as well as consensus on how to define and measure harms in studies examining CRCSPs.

## Supporting information

**S1 File. Supporting Information and Data.** S1 includes Appendices A–I, which contain information and data that support the manuscript.
(DOCX)

**S2 File. All Extracted Outcomes.** S2 includes Appendices J and K, which contain all extracted outcomes from RCTs and SRs, respectively.
(DOCX)

**S3 Checklist. PRIOR Checklist.** S3 includes the PRIOR checklist, which serves as a reporting guideline for overviews of reviews of healthcare interventions, aligned with the content of the manuscript.
(DOCX)

## Author contributions

**Conceptualization:** Anne Katrine Lykke Bie, Frederik Handberg Juul Martiny, John Brandt Brodersen.

**Data curation:** Anne Katrine Lykke Bie, Frederik Handberg Juul Martiny.

**Formal analysis:** Anne Katrine Lykke Bie, John Brandt Brodersen.

**Investigation:** Anne Katrine Lykke Bie.

**Methodology:** Anne Katrine Lykke Bie, Frederik Handberg Juul Martiny, John Brandt Brodersen.

**Project administration:** Anne Katrine Lykke Bie, John Brandt Brodersen.

**Resources:** Anne Katrine Lykke Bie.

**Software:** Anne Katrine Lykke Bie.

**Supervision:** John Brandt Brodersen.

**Validation:** Anne Katrine Lykke Bie, John Brandt Brodersen.

**Visualization:** Anne Katrine Lykke Bie.

**Writing – original draft:** Anne Katrine Lykke Bie.

**Writing – review & editing:** Anne Katrine Lykke Bie, Frederik Handberg Juul Martiny, Christian Patrick Jauernik, Or Joseph Rahbek, Sigrid Brisson Nielsen, Emma Grundtvig Gram, Isabella Kindt, John Brandt Brodersen.

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
