## [Decision Letter · Decision Letter 0]

4 Mar 2025

PONE-D-24-59046Physical Harms in Colorectal Cancer Screening: An Overview of the Reporting in Systematic Reviews and Randomised Controlled TrialsPLOS ONE

Dear Dr. Bie,

Thank you for submitting your manuscript to PLOS ONE. After careful consideration, we feel that it has merit but does not fully meet PLOS ONE’s publication criteria as it currently stands. Therefore, we invite you to submit a revised version of the manuscript that addresses the points raised during the review process.

Dear Respectable AuthorsWe have reached a decision regarding your manuscript based on the reviewers' comments.Please respond to the reviewers' comments as soon as possible and submit the response to the reviewers' file separately. In addition, highlight the changes in the text of the manuscript with yellow highlighter.Our decision is: Major revision

We look forward to receiving your revised manuscript.

Kind regards,

Morteza Arab-Zozani, Ph. D.

Academic Editor

PLOS ONE

2.  As required by our policy on Data Availability, please ensure your manuscript or supplementary information includes the following:

3. Please provide captions for Figure 1, Figure 2 in your manuscript.

4. Please provide captions for Table 6a, 6b, 7 and 8 in your manuscript.

5. We notice that your supplementary tables (appendices) are included in the manuscript file. Please remove them and upload them with the file type 'Supporting Information'. Please ensure that each Supporting Information file has a legend listed in the manuscript after the references list.

Reviewers' comments:

Reviewer's Responses to Questions

**Comments to the Author**

1. Is the manuscript technically sound, and do the data support the conclusions?

Reviewer #1: Yes

Reviewer #2: Yes

2. Has the statistical analysis been performed appropriately and rigorously? 

Reviewer #1: Yes

Reviewer #2: N/A

3. Have the authors made all data underlying the findings in their manuscript fully available?

Reviewer #1: Yes

Reviewer #2: Yes

4. Is the manuscript presented in an intelligible fashion and written in standard English?

Reviewer #1: Yes

Reviewer #2: Yes

5. Review Comments to the Author

Reviewer #1: This is an interesting systematic review exploring the harms related to CRC screening in several countries.

The article is clearly prsenetd and addresses some important issues that warrant its publication.

The authors studied a number of randomized clinical trials (RCTs) and systematic reviews (SRs) on the topic, and presented the coverage of reported harms in those studies. From their study, it appears evident that some RCTs and SRs are under-reporting important harms to individuals undergoing CRC screening. However, an important piece of information not readily presented in the manuscript is the percentage of individuals who suffered those harms. I wonder whether that information could be extracted from the RCTs and SRs and presented. To better understand the "harm of under-reporting harms" (if I may repeat myself), it seems sensible to consider not only the severity of the outcomes but also their incidence.

I would also suggest reformatting tables 2a (clinical trials) and 2b (systematic review), which in the current version are presented divided into several rows with no clear order (neither alphabetical nor chronological). Tables with 5 columns, i.e., Study ID, year, number of patients, protocol, and deviation, would be easier to read and, importantly, to automatically parse for reuse. In these tables, the authors employ "+", "%", and "-" to indicate "yes", "no", and "not relevant", respectively. I do not know if that is some standard notation, but it is certainly very counterintuitive that the opposite of "+" is "%", not "-", and that "-" is employed to indicate "not relevant". I would suggest to revise that notation and, if possible, change it to "+", "-", and "n.r.". Moeover, since there is not strong space limitation in those tables, it might be better not to use symbols at all, and just indicate "yes", "no", and "not relevant" inside the cells.

Reviewer #2: This systematic review provides a valuable and comprehensive analysis of the completeness of reporting of physical harms in randomized controlled trials (RCTs) and systematic reviews (SRs) of colorectal cancer screening programs (CRCSP). The authors compared the levels of harm reporting across studies, utilizing a reference standard encompassing 17 potential types of physical harm. However, several issues need to be addressed by the authors

This article offers a valuable overview of the comprehensiveness of reporting physical harms in colorectal cancer screening programs (CRCSP). However, considering CRCSP's multi-stage nature and the potential for diverse harms at each stage, a schematic diagram integrating the screening process, the scope of this systematic review, and the key questions would significantly enhance the article's clarity. Such a visual representation would provide readers with a more intuitive understanding of the research context and the interconnectedness of its elements. Therefore, I strongly recommend the authors consider incorporating such a diagram.

In the Methods section, it is mentioned that articles in different languages were included; however, this potential language bias is not discussed in the Discussion section. It is recommended that the authors address the potential impact of language bias on the results and evaluate how this limitation might affect the generalizability of the study.

While the authors' attempt to classify harm severity is commendable, there is some ambiguity in the definitions and results that could affect the clarity of the findings. To enhance the reliability of the study, it would be beneficial to provide more precise definitions, ensure consistent severity assessments, and offer more detailed reporting. Additionally, although the authors offer valuable recommendations for improving harm reporting in future studies, these suggestions would benefit from more practical details on their implementation, such as developing a consensus-based typology and optimizing measurement methods. It would be helpful if the authors could further elaborate on these aspects in the Discussion section to strengthen the overall impact of their recommendations.

The authors state that they did not assess the risk of bias (internal validity) as the aim of the study was to assess the quality of reporting and not the validity of study findings. While I understand the authors' intent to focus on reporting quality, the lack of assessment of internal validity is a significant limitation. Internal validity is still important, as studies with high bias may distort reporting quality. A brief bias assessment, even if not the primary focus, would have strengthened the study's conclusions.

The authors have done a good job addressing the inconsistent reporting of harms and how this may lead to an underestimation of their magnitude in relation to CRCSPs, which is an important consideration. The need for better adherence to existing guidelines and international consensus on defining and measuring harms in CRCSP studies is well noted. However, an additional point to consider is the potential for publication bias: Are studies with more severe harms more likely to be published, while those with less severe but still significant harms are overlooked? This aspect could further enhance the understanding of the completeness of harm reporting and should be discussed in more detail.

6. PLOS authors have the option to publish the peer review history of their article (what does this mean? ). If published, this will include your full peer review and any attached files.

**Do you want your identity to be public for this peer review?** For information about this choice, including consent withdrawal, please see our Privacy Policy .

Reviewer #1: No

Reviewer #2: No

---

## [Author Response · Author response to Decision Letter 1]

30 May 2025

Dear EDITOR

Thank you for considering our responses to peer review and our revised manuscript for publication in PLOS One. Below, we respond to your comments about the journal requirement and to the comments from the peer reviewers. We have responded to all reviewer feedback, in bold font below, and revised the manuscript accordingly where relevant.

Additionally, we have looked through the manuscript and the additional files for any remaining typos or grammatical errors, and we have improved the language throughout. All changes are marked in tracked changes in the uploaded manuscript file and in the supplementary files.

Response to reviewers

Reviewer #1

This is an interesting systematic review exploring the harm related to CRC screening in several countries. The article is clearly presented and addresses some important issues that warrant its publication.

We appreciate the reviewer’s careful assessment of our study. While it may appear to be a systematic review, our study is an overview of how physical harms related to CRC screening are reported in systematic reviews and RCTs. We did conduct a systematic search to identify other systematic reviews and we relied on the search strategy from the reference review to identify RCTs. Rather than synthesizing primary data, our focus is on assessing the reporting practices in these sources. Our analysis is based on studies included in our prior systematic review, which has been published in two separate articles covering the risk of deaths and cardiopulmonary events and the risk of bleeding and perforation of CRC screening[1, 2]. We have clarified this distinction in the revised manuscript to ensure that the scope and purpose of our study are accurately conveyed.

The authors studied a number of randomized clinical trials (RCTs) and systematic reviews (SRs) on the topic, and presented the coverage of reported harms in those studies. From their study, it appears evident that some RCTs and SRs are under-reporting important harms to individuals undergoing CRC screening. However, an important piece of information not readily presented in the manuscript is the percentage of individuals who suffered those harms. I wonder whether that information could be extracted from the RCTs and SRs and presented. To better understand the "harm of under-reporting harms" (if I may repeat myself), it seems sensible to consider not only the severity of the outcomes but also their incidence.

Thank you for your comment. It is correct that this study assessed how many of the 17 potential types of harm identified in the reference review that were reported per study (study coverage) and across studies for each of the 17 types of outcomes (outcome coverage). Indeed, it was the aim of the reference review (Martiny et al.) to assess the types of physical harm related to CRC screening and to quantify the risk of each of these harms to the extent possible, i.e., the incidence of harms requested by the peer reviewer. In the reference review, we found that there were 17 distinct types of physical harms related to CRC screening. We have reported the risk of deaths and cardiopulmonary events and the risk of bleeding and perforation during CRC screening in two separate articles (Martiny et al. and Kindt et al.). Regrettably, the heterogeneity related to outcome definition, measurement and reporting for the remaining 13 types of physical harm hindered any meaningful quantification of these outcomes. Because it was not possible to report the incidence of these harms as suggested, but we noted much heterogeneity about the types of harms studied and reported across RCT and SRs, we chose to conduct the present study to to examine the difference between the reporting of the 17 types of harm in RCTs and SRs that investigated physical harm related to CRC screening. Still, the number of people participating in RCT of CRCSPs is noted in Table 1a, giving an idea of the number of people who might have experienced harms that were not reported. Yet, we consider it outside the aim of the article to speculate how many of these people might have experienced harms not reported in studies. However, we do appreciate the comment and have worked with the language and definitions of research questions so that the aims of this study are hopefully clearer.

I would also suggest reformatting tables 2a (clinical trials) and 2b (systematic review), which in the current version are presented divided into several rows with no clear order (neither alphabetical nor chronological).

Tables with 5 columns, i.e., Study ID, year, number of patients, protocol, and deviation, would be easier to read and, importantly, to automatically parse for reuse. In these tables, the authors employ "+", "%", and "-" to indicate "yes", "no", and "not relevant", respectively. I do not know if that is some standard notation, but it is certainly very counterintuitive that the opposite of "+" is "%", not "-", and that "-" is employed to indicate "not relevant". I would suggest to revise that notation and, if possible, change it to "+", "-", and "n.r.". Moreover, since there is not strong space limitation in those tables, it might be better not to use symbols at all, and just indicate "yes", "no", and "not relevant" inside the cells.

Thank you for these relevant comments. The content from table 2a and 2b has been added to table 1 under the suggested headings. Further, we have changed the terminology to “yes”, “no”, and “Not relevant” to improve readability and reusability as pointed out.

Reviewer #2

This systematic review provides a valuable and comprehensive analysis of the completeness of reporting of physical harms in randomized controlled trials (RCTs) and systematic reviews (SRs) of colorectal cancer screening programs (CRCSP). The authors compared the levels of harm reporting across studies, utilizing a reference standard encompassing 17 potential types of physical harm. However, several issues need to be addressed by the authors.

Thank you for your time and consideration reviewing our manuscript. We have addressed each of your comments below and believe your suggested edits have improved the quality of our reporting.

This article offers a valuable overview of the comprehensiveness of reporting physical harms in colorectal cancer screening programs (CRCSP). However, considering CRCSP's multi-stage nature and the potential for diverse harms at each stage, a schematic diagram integrating the screening process, the scope of this systematic review, and the key questions would significantly enhance the article's clarity. Such a visual representation would provide readers with a more intuitive understanding of the research context and the interconnectedness of its elements. Therefore, I strongly recommend the authors consider incorporating such a diagram.

We sincerely appreciate the reviewer’s thoughtful suggestion. We agree that clarifying which steps of the screening cascade we assess and incorporating a schematic diagram would enhance the readability and clarity of the manuscript. Therefore, we have explicitly stated the specific steps of the screening cascade that our study focuses on and have added an illustration of the screening process, please see figure 1, page 4, line 15 (uploaded as separate files). We believe this addition provides a clearer visual representation of the research context and improves the overall accessibility of our findings. Thank you for this valuable recommendation.

In the Methods section, it is mentioned that articles in different languages were included; however, this potential language bias is not discussed in the Discussion section. It is recommended that the authors address the potential impact of language bias on the results and evaluate how this limitation might affect the generalizability of the study.

Thank you for your valuable comment. We agree that discussing potential language bias is important for assessing the generalizability of our findings. In both the reference review and this overview of the reporting of physical harm in RCTs and SRs on CRC screening, we did not impose language restrictions on the primary articles. However, among the nine SRs included in our overview, one only included studies reported in English, one included studies in English or French, four had no language restrictions, and one did not report language limitations. While most studies are typically published in English, some may have been missed, and the predominance of studies from Western or developed countries also impacts generalizability. This introduces a possible language bias, as relevant non-English studies may have been overlooked, potentially affecting the comprehensiveness of the evidence. We have now explicitly acknowledged these concerns in the limitations section of the discussion, see page 26, line 25-31.

While the authors' attempt to classify harm severity is commendable, there is some ambiguity in the definitions and results that could affect the clarity of the findings. To enhance the reliability of the study, it would be beneficial to provide more precise definitions, ensure consistent severity assessments, and offer more detailed reporting.

On page 8, line 7-18, in the Methods section about research question 5 severity assessment, we have written:

“We tried to identify an existing classification system of complications but found none that were appropriate to use on complications from CRCSPs. For that reason, we developed a severity assessment: for all types of harm, we distinguished between five potential severity categories: Not Severe, Severe, Very Severe, Death, and Unknown. We then categorized each type of harm reported in RCTs and SRs according to these five categories. We used information from studies to categorize the severity of the type of harm. When an outcome had a fatal outcome, it was categorized as “Death”. When harms were vaguely defined or when consequences of the type of harm were not reported, we categorized the harm assessment as “Unknown”. We distinguished between “Severe” and “Very severe”, categorizing outcomes as the latter when they led to further surgical procedures or prolonged hospitalization. The fifth category, “Not severe” was used when authors described the outcome as not severe or specified that the outcome led to a minimum of discomfort. The developed rules of categorisation can be seen in Appendix E.”

In the methods section we noted that we were unable to identify a sufficiently comprehensive existing classification system to assess the severity of physical harm related to CRC screening. Consequently, we developed a five-category classification, which is further detailed in Appendix E. However, we acknowledge that this classification is imperfect and that we faced challenges in distinguishing subtle differences in the severity of harm within these categories. To emphasise the lack of a standardised classification system—which contributes to significant heterogeneity in the measurement and reporting of harm severity—and to acknowledge the limitations of our approach, we have now addressed this issue in the discussion section.

Additionally, although the authors offer valuable recommendations for improving harm reporting in future studies, these suggestions would benefit from more practical details on their implementation, such as developing a consensus-based typology and optimizing measurement methods. It would be helpful if the authors could further elaborate on these aspects in the Discussion section to strengthen the overall impact of their recommendations.

Thank you for your comment. We have elaborated on our suggestions for developing a consensus-based typology for the reporting of harms, and optimising measurement methods in the discussion in the section about future research, page 29, line 30 – page 30, line 5.

The authors state that they did not assess the risk of bias (internal validity) as the aim of the study was to assess the quality of reporting and not the validity of study findings. While I understand the authors' intent to focus on reporting quality, the lack of assessment of internal validity is a significant limitation. Internal validity is still important, as studies with high bias may distort reporting quality. A brief bias assessment, even if not the primary focus, would have strengthened the study's conclusions.

In the reference review that aimed to quantify the risk of physical harm related to CRC screening programmes, we assessed the risk of bias on the risk estimates, i.e., the internal validity of harm estimates from original studies. An assessment of bias is an assessment of the internal validity of studies regarding whether the results of those studies might overestimate or underestimate the true intervention effect[3] due to various sources of bias.

However, this overview specifically focuses on the quality of reporting in RCTs and SRs of physical harms, which is not related to the potential biases due to study design, measurement error, or other types of biases that may affect harm estimates.

Still, our general findings about the low study coverage of physical harms in most RCTs and SRs and the low outcome coverage for many types of physical harm could be seen as sign of non-reporting bias, also termed selective reporting bias, i.e., when decisions about how, when and where to report results in primary studies or in reviews of those studies are influenced by the nature and direction of the results. Further, there is always a risk of non-reporting bias where only certain outcomes or analyses within a study are reported, often favouring positive or favourable results. In other words, outcomes may not have been reported even if they occurred or might not have been reported because they did not occur. It was beyond the scope of this paper to assess the risk of these biases as we did in the reference review. However, we have written about the importance of reporting harms even if they do not occur on page 27, line 12 - page 28, line 6.

These concerns are important to understand the implications of our findings. Therefore, we have added these reflections to the discussion, page 27, line 13 – page 28, line 13.

The authors have done a good job addressing the inconsistent reporting of harms and how this may lead to an underestimation of their magnitude in relation to CRCSPs, which is an important consideration. The need for better adherence to existing guidelines and international consensus on defining and measuring harms in CRCSP studies is well noted. However, an additional point to consider is the potential for publication bias: Are studies with more severe harms more likely to be published, while those with less severe but still significant harms are overlooked? This aspect could further enhance the understanding of the completeness of harm reporting and should be discussed in more detail.

Thank you for your very relevant comment. Please see the answer above.

References

1. Martiny F, Bie A, Jauernik C, Rahbeck O, Nielsen SB, Brodersen J. Physical harms associated with colorectal cancer screening’s diagnostic work-up phase: a systematic review and meta-analysis of the risk of death and cardiopulmonary events. Paper in Progress.

2. Kindt IS, Martiny FHJ, Gram EG, Bie AKL, Jauernik CP, Rahbek OJ, et al. The risk of bleeding and perforation from sigmoidoscopy or colonoscopy in colorectal cancer screening: A systematic review and meta-analyses. PLoS One. 2023;18(10):e0292797. Epub 2023/10/31. doi: 10.1371/journal.pone.0292797. PubMed PMID: 37906565; PubMed Central PMCID: PMCPMC10617695 to PLOS ONE policies on sharing data and materials.

3. Cochrane library https://training.cochrane.org/handbook/current/chapter-07.

---

## [Decision Letter · Decision Letter 1]

23 Jun 2025

PONE-D-24-59046R1Physical Harms in Colorectal Cancer Screening: An Overview of the Reporting in Systematic Reviews and Randomised Controlled TrialsPLOS ONE

Dear Dr. Bie,

Thank you for submitting your manuscript to PLOS ONE. After careful consideration, we feel that it has merit but does not fully meet PLOS ONE’s publication criteria as it currently stands. Therefore, we invite you to submit a revised version of the manuscript that addresses the points raised during the review process.

We look forward to receiving your revised manuscript.

Kind regards,

Morteza Arab-Zozani, Ph. D.

Academic Editor

PLOS ONE

Journal Requirements:

Reviewers' comments:

Reviewer's Responses to Questions

**Comments to the Author**

1. If the authors have adequately addressed your comments raised in a previous round of review and you feel that this manuscript is now acceptable for publication, you may indicate that here to bypass the “Comments to the Author” section, enter your conflict of interest statement in the “Confidential to Editor” section, and submit your "Accept" recommendation.

Reviewer #1: All comments have been addressed

Reviewer #2: All comments have been addressed

2. Is the manuscript technically sound, and do the data support the conclusions?

Reviewer #1: Yes

Reviewer #2: Yes

3. Has the statistical analysis been performed appropriately and rigorously? 

Reviewer #1: Yes

Reviewer #2: N/A

4. Have the authors made all data underlying the findings in their manuscript fully available?

Reviewer #1: Yes

Reviewer #2: Yes

5. Is the manuscript presented in an intelligible fashion and written in standard English?

Reviewer #1: Yes

Reviewer #2: Yes

6. Review Comments to the Author

Reviewer #1: I had just two comments on the original version of the article.

The authors addressed one of them, regarding the formatting of some tables.

The other comment was not fully addressed (the incidence of the unreported harms) but the authors provided a sensible reason for not doing so.

I think the article is ready for publication.

Reviewer #2: Thank you for the revised version of the manuscript. The authors have addressed the previous comments in a satisfactory manner. Overall, the manuscript has improved in both content and presentation. I believe it is suitable for publication after minor revision.

In line 10, the quotation mark (") at the beginning of the paragraph appears to be unnecessary or mistakenly placed, as it is not closed later in the text. The authors are kindly requested to check the manuscript for clarity, consistency, and minor formatting issues.

Implications for screening participants

10 "We believe it is important…..

7. PLOS authors have the option to publish the peer review history of their article (what does this mean? ). If published, this will include your full peer review and any attached files.

**Do you want your identity to be public for this peer review?** For information about this choice, including consent withdrawal, please see our Privacy Policy .

Reviewer #1: No

Reviewer #2: No

---

## [Author Response · Author response to Decision Letter 2]

23 Jun 2025

Thank you for considering our responses to peer review and our revised manuscript for publication in PLOS One. Below, we respond to your comments about the journal requirement and to the comments from the peer reviewers. We have responded to all reviewer feedback, in bold font below, and revised the manuscript accordingly where relevant.

Response to reviewers

6. Review Comments to the Author

Reviewer #1: I had just two comments on the original version of the article.

The authors addressed one of them, regarding the formatting of some tables.

The other comment was not fully addressed (the incidence of the unreported harms) but the authors provided a sensible reason for not doing so.

I think the article is ready for publication.

Thank you so much for your valued comments.

Reviewer #2: Thank you for the revised version of the manuscript. The authors have addressed the previous comments in a satisfactory manner. Overall, the manuscript has improved in both content and presentation. I believe it is suitable for publication after minor revision.

In line 10, the quotation mark (") at the beginning of the paragraph appears to be unnecessary or mistakenly placed, as it is not closed later in the text. The authors are kindly requested to check the manuscript for clarity, consistency, and minor formatting issues.

Implications for screening participants

10 "We believe it is important…..

We thank the reviewer for their careful and attentive reading of the manuscript. The typographical error has been corrected as suggested. Additionally, we have reviewed the manuscript and the additional files for any remaining typos or grammatical errors. All changes are marked in tracked changes in the uploaded manuscript file.

---

## [Decision Letter · Decision Letter 2]

12 Aug 2025

Physical Harms in Colorectal Cancer Screening: An Overview of the Reporting in Systematic Reviews and Randomised Controlled Trials

PONE-D-24-59046R2

Dear Dr. Bie,

We’re pleased to inform you that your manuscript has been judged scientifically suitable for publication and will be formally accepted for publication once it meets all outstanding technical requirements.

Kind regards,

Zubing Mei, MD,PH.D

Academic Editor

PLOS ONE

Additional Editor Comments (optional):

Reviewers' comments:

Reviewer's Responses to Questions

**Comments to the Author**

1. If the authors have adequately addressed your comments raised in a previous round of review and you feel that this manuscript is now acceptable for publication, you may indicate that here to bypass the “Comments to the Author” section, enter your conflict of interest statement in the “Confidential to Editor” section, and submit your "Accept" recommendation.

Reviewer #1: All comments have been addressed

2. Is the manuscript technically sound, and do the data support the conclusions?

Reviewer #1: Yes

3. Has the statistical analysis been performed appropriately and rigorously? 

Reviewer #1: Yes

4. Have the authors made all data underlying the findings in their manuscript fully available?

Reviewer #1: Yes

5. Is the manuscript presented in an intelligible fashion and written in standard English?

Reviewer #1: Yes

6. Review Comments to the Author

Reviewer #1: My comments were already addressed in a previous revision.

As stated, one of the comments was fully addressed, and the authors provided a sensible reason for not addressing the second one.

7. PLOS authors have the option to publish the peer review history of their article (what does this mean? ). If published, this will include your full peer review and any attached files.

**Do you want your identity to be public for this peer review?** For information about this choice, including consent withdrawal, please see our Privacy Policy .

Reviewer #1: No

---

## [Editor Report · Acceptance letter]

PONE-D-24-59046R2

PLOS ONE

Dear Dr. Bie,

I'm pleased to inform you that your manuscript has been deemed suitable for publication in PLOS ONE. Congratulations! Your manuscript is now being handed over to our production team.

Kind regards,

on behalf of

Dr. Zubing Mei

Academic Editor

PLOS ONE